# Effect of Long-Term Proton Pump Inhibitor Use on Blood Vitamins and Minerals: A Primary Care Setting Study

**DOI:** 10.3390/jcm12082910

**Published:** 2023-04-17

**Authors:** Giuseppe Losurdo, Natale Lino Bruno Caccavo, Giuseppe Indellicati, Francesca Celiberto, Enzo Ierardi, Michele Barone, Alfredo Di Leo

**Affiliations:** 1Section of Gastroenterology, Department of Precision Medicine and Jonic Area, University “Aldo Moro” of Bari, 70124 Bari, Italy; 2Ph.D. Course in Organs and Tissues Transplantation and Cellular Therapies, Department of Precision Medicine Jonic Area, University “Aldo Moro” of Bari, 70124 Bari, Italy; 3College of General Practitionners, ASL BA, 70056 Molfetta, Italy

**Keywords:** proton pump inhibitors, vitamin B12, vitamin D, *Helicobacter pylori*

## Abstract

Background and objectives. Long-term proton pump inhibitor (PPI) use is frequently encountered in primary care. Its effect on micronutrient absorption is known, as vitamin B12, calcium or vitamin D insufficiency may occur in such patients. Materials and methods. We recruited patients using a PPI (pantoprazole) for >12 months. The control group was represented by subjects attending the general practitioner not taking any PPI in the last 12 months. We excluded subjects using nutritional supplements or with diseases interfering with micronutrient blood levels. All subjects underwent blood sampling with full blood count, iron, ferritin, vitamin D, calcium, sodium, potassium, phosphate, zinc and folate. Results. We recruited 66 subjects: 30 in the PPI group and 36 in the control group. Long-term pantoprazole users had lower red blood cell count but similar hemoglobin. We did not find any significant difference in blood iron, ferritin, vitamin B12 and folate. Vitamin D deficit was observed more frequently in the PPI group (100%) than in controls (30%, *p* < 0.001), with blood levels lower in pantoprazole consumers. No differences in calcium, sodium and magnesium were observed. Pantoprazole users had lower phosphate levels than controls. Finally, a non-significant trend for zinc deficiency was found in PPI users. Conclusions. Our study confirms that chronic PPI users may encounter alterations in some micronutrients involved in bone mineral homeostasis. The effect on zinc levels deserves further investigation.

## 1. Introduction

Proton Pump Inhibitors (PPIs) are a class of widely used medications with different indications. In Italy, they are currently approved by the “Agenzia Italiana del Farmaco” (AIFA, i.e., the Italian drug administration agency) for the management of peptic ulcer disease, esophageal reflux disease with or without esophagitis, Zollinger–Ellison Syndrome, non-ulcerative dyspepsia, as well as for the prevention of gastrointestinal ulcers in conditions, such as chronic non-steroidal anti-inflammatory drug (NSAID) administration [1]. PPIs are also crucial in treating *Helicobacter pylori* infection in association with antibiotics [2]. Their mechanism of action is the inhibition of the H+,K+ ATPase located in gastric parietal cells [3].

Long-term use of these medications has become extremely common, and some adverse effects have been observed linked to their chronic use [4]. These adverse effects can be divided into: (i) infective, (ii) related to hypergastrinemia and (iii) absorption deficiencies due to hypochlorhydria [5]. Infections recognize the increase in gastric pH as a possible mechanism, i.e., the reduction in a first-line barrier against bacteria introduced with food [5]. The most common infections associated with PPI chronic use are C. difficile infection and non-typhoid Salmonella and Campylobacter enteric infections; studies have shown conflicting results for other infections, such as spontaneous bacterial peritonitis, encephalopathy in cirrhotic patients and community-acquired pneumonia [5]. The most important effects related to hypergastrinemia are gastric hyperplasia/metaplasia and rebound acid hypersecretion [5]. The most-studied absorption deficiencies associated with chronic PPI administration are vitamin B12 and iron deficiency (anemia), hypomagnesemia and alterations in vitamin D and calcium metabolism, with some studies showing an increased bone fracture risk [5,6]; in particular, a meta-analysis of 18 studies showed that PPI use could moderately increase the risk of hip fracture with an RR = 1.26 [7]. Other potential adverse effects with weaker evidence are dementia and renal disease [6].

Several studies [8,9,10,11] assessed the importance of gastric acid secretion in vitamin and mineral absorption. For example, gastric pH contributes to reducing ferric ion into its ferrous form, which is more soluble, thus avoiding the formation of ferric complexes that reduce absorption and bioavailability [9]. Given that defective intestinal calcium absorption has been reported in patients with gastric achlorhydria, it is thought that the pH-dependent ionization of calcium salts from ingested foods is a prerequisite for its absorption [10]. Regarding vitamin B12, several factors are involved in its absorption. First, peptic enzymes and the gastric acidic environment contribute to the release of B12 from food; then, the gastric pH favors the binding between vitamin B12 and Haptocorrin (HC), a protein secreted by the salivary glands [12]. This prevents vitamin B12 hydrolysis by gastric acid and reduces degradation by intestinal microbiota [12]. In the duodenum, pH elevation decreases the affinity between vitamin B12 and HC, promoting B12 binding to another protein, the intrinsic factor (IF) [12]. IF is secreted by gastric mucosa and helps B12 absorption through the intestinal wall in the distal ileum by the binding to Cubulin, the receptor of complex B12-IF [12].

It is evident that gastric acid has a role in absorption mechanisms and, thus, PPIs, by blocking acid gastric secretion, which may likely interfere with these absorption mechanisms. In this prospective case–control study, we aimed to investigate possible alterations in micronutrients in chronic PPI users in a general practitioner setting.

## 2. Materials and Methods

The study was planned as a transversal case–control study. We recruited, in the period January–December 2020, patients using PPIs (pantoprazole) for at least 12 months attending the general practitioner. Control group was represented by subjects not taking any PPI in the last 12 months, recruited in the same setting. We excluded subjects < 18 years old. We excluded subjects taking oral or parenteral nutritional supplements containing iron, folate, vitamin B12 or D, or taking other medications (such as corticosteroids), which could influence vitamin and micronutrient levels. Additionally, we ruled out subjects with diseases (atrophic gastritis, celiac disease, inflammatory bowel disease, and hematological disease) that could cause alterations in micronutrient blood levels. All subjects underwent blood sampling with full blood count, iron, ferritin, vitamin D, calcium, sodium, potassium, phosphate, zinc and folate. The study was carried out in accordance with the Declaration of Helsinki and approved by the Internal Gastroenterology Institutional Review Board of our unit. Written informed consent was obtained from all subjects involved in the study. All laboratory analyses were performed according to the local best-practice rule. Considering the prevalence of vitamin D deficiency, by estimating a difference in prevalence of 35% between the two groups (which is much less than what we found, i.e., 75%), an alpha error of 0.05 and a statistical power of 80%, the total sample size would be 60 patients (30 per group).

Student’s t test and Fisher’s exact test were used for statistical comparison for continuous and dichotomous variables, respectively. Correlation analysis was performed via Spearman r coefficient. A multivariate binomial regression analysis was used to assess odds ratios (ORs) and relative 95% confidence intervals (95% CIs). The statistical software programs GraphPad Prism version 5 (San Diego, CA, USA) and SPSS 23 were used.

## 3. Results

We recruited 66 subjects: 32 males and 34 females; there were 30 in the PPI group and 36 in the control group. In detail, all patients assumed pantoprazole at a dose of 40 mg/day. Patients in the PPI group were older than controls (75.6 ± 9.6 versus 63.2 ± 16.4 years, *p* < 0.001). Regarding comorbidities, the most common ones in the PPI group were hypertension (n = 26) and diabetes (n = 6).

Long-term PPI users had lower red blood cell count (4.26 ± 0.52 versus 4.86 ± 0.36 × 106, *p* < 0.001) but similar hemoglobin and mean cell volume values. We did not find any significant difference in blood iron (85.9 ± 22.3 versus 83.4 ± 32.3, *p* = 0.71), ferritin (170.0 ± 271.0 versus 364.5 ± 677.1, *p* = 0.14), vitamin B12 (441.2 ± 195.5 versus 419.4 ± 227.3, *p* = 0.68) and folate (8.5 ± 3.5 versus 8.6 ± 2.9, *p* = 0.92).

Vitamin D deficiency was observed more frequently in the PPI group (100%) than in controls (25%, *p* < 0.001), with blood levels lower in PPI consumers (15.5 ± 6.8 versus 36.6 ± 21.2, *p* < 0.001). No differences in calcium and magnesium levels were observed. PPI users had lower phosphate (3.44 ± 0.60 versus 3.98 ± 0.73, *p* = 0.002).

Regarding other minerals, potassium levels were lower than controls (4.18 ± 0.47 versus 4.49 ± 0.54, *p* = 0.02). Finally, a non-significant trend for zinc deficiency was found in PPI users (81.2 ± 13.9 versus 87.0 ± 12.7, *p*= 0.08) and also a borderline significant trend for lower sodium levels in PPI users (139.8 ± 3.8 versus 141.8 ± 4.6, *p* = 0.05).

The overall results are summarized in Table 1.

To exclude the effect of age as a confounding factor for vitamins and minerals involved in bone metabolism, we correlated age with such micronutrients and found no correlation with vitamin D (r = 0.08, *p* = 0.64), calcium (r = 0.25, *p* = 0.18), phosphate (r = −0.22, *p* = 0.23) and zinc (r = −0.23, *p* = 0.21).

Furthermore, a multivariate analysis aiming to explore the possible associated and confounding factors for vitamin D deficiency, reported in Table 2, did not find any variable associated with such a deficit.

## 4. Discussion

Several studies investigated the link between chronic administration of PPIs and vitamins and mineral absorption. According to our results, the most important studies, which evaluated long-term use of PPIs on B12 absorption [13,14,15,16], concluded that there were no significant alterations in vitamin B12 levels during long-term PPI therapy, except in the case of Zollinger–Ellison syndrome patients, which were completely achlorhydric due to heavy PPI treatment, with a consequent decrease in vitamin B12 blood levels [14]. However, the literature is quite heterogeneous when dealing with PPI-induced adverse events. For example, we added a study that found that PPI consumers had slightly higher vitamin B12 [17], which is in full disagreement with another that did not find any effect on folate, B12, calcium and vitamin D [18].

There are some case reports [19] and a review [20] that correlate the chronic use of PPIs to hypomagnesemia, with a pooled relative risk of hypomagnesemia in patients with PPIs of 1.43 (95% CI, 1.08–1.88) [18]. The proposed mechanism [21] involves an increase in the pH in the intestinal lumen caused by PPI chronic administration, which reduces the affinity between the Mg+ ion and its transporter, Transient Receptor Potential Melastatin 6/7 (TRPM 6/7). This reduction in binding between Mg+ ion and its transporter may trigger mRNA transcription of TRPM 6 in most individuals, but epigenetic modifications may explain why hypomagnesemia occurs only in a few individuals [21]. In our study, there was no significative difference in magnesium levels between the two groups (Table 1), maybe due to the small sample size.

The acidic pH of gastric juice could affect the bioavailability of vitamin D, despite the fact that there are no available data on the susceptibility of major dietary forms of vitamin D to gastrointestinal pH conditions [22]. Another mechanism that could explain the influence of PPIs on vitamin D homeostasis is through hypomagnesemia because several steps in vitamin D metabolism depend on magnesium as a cofactor, such as vitamin D binding to vitamin D binding protein, 25(OH)D synthesis, 1,25 (OH)2D synthesis, 25-hydroxylase synthesis and vitamin D receptor expression for cellular effects [23]. Sharara et al. [24] analyzed the effect of PPIs on bone metabolism, including vitamin D. Most of the study participants had hypovitaminosis D at baseline, but vitamin D increased in both the PPI and control groups after 3 months of treatment without statistically significative difference between the two groups (*p* = 0.971). In another study, by Hinson et al. [25], which evaluated the hyperparathyroidism associated with PPI therapy, the differences in vitamin D levels in patients taking PPIs for at least 6 months and in patients without PPIs were not statistically significant, independently from bisphosphonate co-administration. These findings do not seem to agree with our findings: significantly lower vitamin D levels in PPI consumers (*p* < 0.001). This could be explained by the fact that the PPI-user sample is significantly older (*p* < 0.001) than the non-PPI-user group in our study, as advanced age is a factor associated with suboptimal vitamin D levels [26]. Other factors that could explain our result could be the fact that body mass index was not considered in this evaluation, since 25 (OH) vitamin D, which is fat-soluble, can be sequestered in the adipose tissue [27]. Additionally, specific interfering drugs were not considered in our study design (for example, bile acid sequestrants, anti-epileptic drugs) [28]. Finally, we selected subjects taking pantoprazole for >12 months, a longer time from the 6 months of Hinson.

Theoretically, an acidic environment in the stomach facilitates the release of ionized calcium from insoluble calcium salts such as calcium carbonate [29]. In fact, in achlorhydric patients, the absorption of insoluble calcium salts such as calcium carbonate taken under fasting conditions virtually does not occur, while soluble calcium salts such as calcium citrate are still normally absorbed [29]. Despite these results, a review by Insogna et al. [30] analyzed several studies about the influence of PPIs on calcium absorption, finding conflicting results. Moreover, most of the analyzed studies were on patients with comorbidities that could influence calcium metabolism (chronic kidney disease or achlorhydria). Thus, it is not possible to state the effective role of PPIs on calcium intestinal absorption. On the other hand, it has been demonstrated that PPIs can also inhibit osteoclast proton pumps, reducing bone calcium phosphate reabsorption, thus decreasing calcium blood levels [31]. Our study did not show any significant differences between calcium levels in PPIs users versus controls (Table 1). In this regard, the American Gastroenterology Association guidelines [32] do not support either the routine supplementation of calcium intake in chronic PPI users, unless it is below the recommended daily allowance, independently of PPI administration, or the routine evaluation of bone mass density in such patients.

Termanini et al. [15], in his study on patients with Zollinger–Ellison syndrome, investigated, apart from B12 levels, folate levels and complete blood counts yearly for a mean of 4.5 years. He did not find any significant decrease in these parameters. Another study, by Attwood et al. [33], who evaluated the safety of long-term pantoprazole therapy by comparing the results of the LOTUS and the SOPRAN studies (two studies that compared PPI therapy vs. surgery in GERD), demonstrated that folate levels did not vary significantly in time in both groups (those receiving PPI therapy and those receiving surgery). These findings are coherent with our findings on folate levels.

Gastric acid facilitates the absorption of non-heme iron by reducing ferric iron to the more soluble ferrous form and also enhances iron salt dissociation from ingested food and allows for the formation of complexes with amines and sugars that also increase absorption [4]. Thus, it is expected that PPIs, by reducing gastric acid secretion, can potentially cause iron malabsorption. Two relatively recent case–control studies [34,35] investigated the link between chronic PPI administration and iron deficiency, finding a positive association between the two conditions, with a stronger association for higher daily doses (>1.5 vs. <0.75 PPI pills/d; *p* value interaction = 0.004), and decreased after medication discontinuation (*p*-trend < 0.001) [34]. Despite these results, a review by Priyanka et al. [36] concluded that “clinically significant iron deficiency is less likely to occur with the long-term use of PPIs in normal subjects although it may happen”. However, PPI use may be associated with difficulty in achieving adequate iron store repletion in iron-deficient subjects. In our study, there was no significant difference in blood iron between subjects taking PPIs and the control group (Table 1).

Finally, we found a non-significant trend for zinc deficiency in PPI users (*p* = 0.08) (Table 1). A study by Farrell et al. [37] evaluated the effect of PPIs on zinc absorption and storage; they found significantly lower levels of blood zinc in chronic PPI users vs. controls, despite the low numerosity in the samples (75 ± 3 mcg/dL vs. 91 ± 3 mcg/dL, *p* = 0.004). Surprisingly, this issue has not been investigated extensively in the literature; therefore, our paper might have merit to have shed new light on the topic, deserving further investigation.

Our study suffers from some limitations, such as the small sample and the missed evaluation of potentially confounding factors including eating habits. However, this was a pilot study performed on a few patients under the care of a single physician; therefore, larger samples are necessary. Furthermore, our study did not evaluate baseline levels of micronutrients, and a study with a follow-up before and after long-term use of PPIs could have been more proper and provided more interesting results. Some imbalances in group composition could be another limitation; for example, the age in the PPI group was higher than in controls. However, the fact that PPI-taking subjects were older than controls may be explained by the fact that these drugs are mostly needed by elderly patients in the long term. However, we found that age did not correlate with any of the levels of micronutrients involved in bone metabolism (Vitamin D, Ca, Zn and P), as shown in Figure 1. This could provide evidence that the use of PPIs could play a role in the deficiency of such molecules more than the age itself [38]. The primary care context is, however, a field in which chronic PPI effects on micronutrients have been poorly explored so far. Sarzynski et al. [39] performed a similar case–control study in an academic outpatient population, but it was retrospective and evaluated only blood markers related to iron deficiency (hemoglobin, hematocrit, mean corpuscular volume). Qorraj-Bytyqi et al. [40] prospectively analyzed vitamin B12, homocysteine and ferritin levels in a population of patients with osteoarthritis undergoing 12 months of PPI therapy associated with NSAIDs at baseline and after 12 months. Thus, other existing studies have some differences from our study, and we did not find other similar studies in general practitioner settings. Moreover, the several existing studies come to different conclusions, and the most recent guidelines [32] do not recommend routine screening or monitoring of bone mineral density, serum creatinine, magnesium or vitamin B12. Moreover, the results of our study and the evidence from the literature should be interpreted with wit, and, when PPI indication is evidence-based, there is no reason for PPI withdrawal because of fear of side effects [41,42]. Of note, ten patients showed APCA positivity, which did not reflect the presence of atrophy at gastric biopsy; such patients could be considered as potential autoimmune gastritis cases [43], but the absence of atrophy warrants that impaired absorption is secondary to pantoprazole and not atrophic gastritis.

## 5. Conclusions

Despite the above-cited limitations, our study seems to confirm that chronic pantoprazole use may engender alterations in micronutrients involved in bone mineral homeostasis, while other element absorption, such as iron and B12, seem not to be affected. Interestingly, the effect on zinc levels is an interesting finding, which requires further investigation.

## Figures and Tables

**Figure 1 jcm-12-02910-f001:**
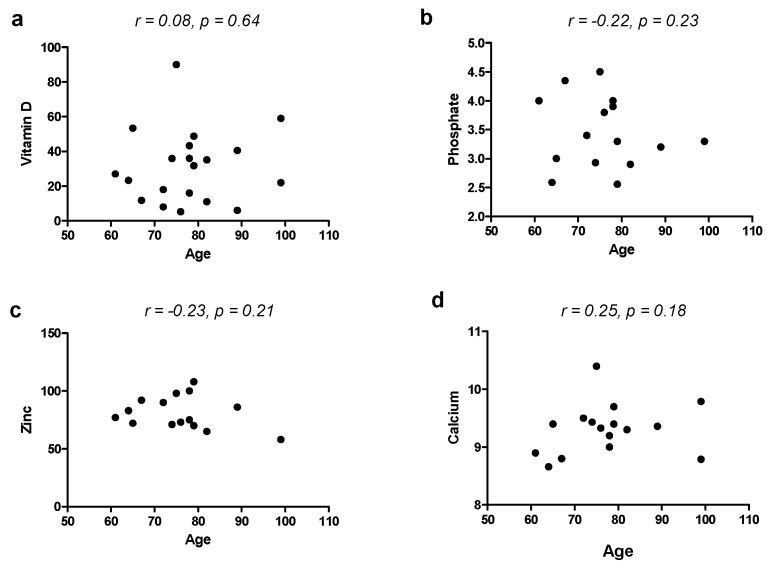
Correlation between age and blood levels of vitamin D (**a**), phosphate (**b**), zinc (**c**) and calcium (**d**).

**Table 1 jcm-12-02910-t001:** Characteristics and laboratory values of the studied population (mean ± SD or percentage).

Variables	PPI Group (n = 30)	Controls (n = 36)	*p* Value
**Age (y)**	75.6 ± 9.6	63.2 ± 16.4	<0.001
**RBC (×10^6^/mm^3^)**	4.26 ± 0.52	4.86 ± 0.36	<0.001
**Hb (g/dL)**	12.6 ± 1.9	13.2 ± 2.3	0.28
**Mean corpuscolar volume (fL)**	88.8 ± 6.3	86.8 ± 1.8	0.28
**WBC (×10^3^/mm^3^)**	6.39 ± 2.02	7.27 ± 2.25	0.99
**Iron (μg/dL)**	85.9 ± 22.3	83.4 ± 32.3	0.71
**Ferritin (μg/L)**	170.0 ± 271.0	364.5 ± 677.1	0.14
**Vitamin D (ng/mL)**	15.5 ± 6.8	36.6 ± 21.2	<0.001
**Vitamin D deficit**	30 (100%)	9 (25%)	<0.001
**Calcium (mg/dL)**	8.7 ± 2.2	9.2 ± 0.4	0.25
**Magnesium (mg/dL)**	2.09 ± 0.33	2.05 ± 0.28	0.58
**Phosphate (mg/dL)**	3.44 ± 0.60	3.98 ± 0.73	0.002
**Zinc (μg/dL)**	81.2 ± 13.9	87.0 ± 12.7	0.08
**Sodium (mEq/L)**	139.8 ± 3.8	141.8 ± 4.6	0.05
**Potassium (mEq/L)**	4.18 ± 0.47	4.49 ± 0.54	0.02
**Gastrin (pg/mL)**	28.9 ± 37.9	39.6 ± 108.5	0.61
**APCA positivity**	4 (13.3%)	6 (16.7%)	0.74
**Vitamin B12 (pg/mL)**	441.2 ± 195.5	419.4 ± 227.3	0.68
**Vitamin B12 deficit**	0 (0%)	4 (11.1%)	0.12
**Folate (ng/mL)**	8.5 ± 3.5	8.6 ± 2.9	0.92
**Folate deficit**	2 (6.7%)	2 (5.5%)	1
**TSH (mU/L)**	2.08 ± 0.86	2.36 ± 0.95	0.21
**TPO (UI/mL)**	7.09 ± 96.3	96.3 ± 346.4	0.16

**Table 2 jcm-12-02910-t002:** Multivariate analysis of factors associated with vitamin D deficiency.

	OR	95% CI	*p* Value
**Calcium**	0.012	0–∞	0.971
**Magnesium**	0.021	0–∞	0.967
**Phosphate**	2577	0.314–639	0.999
**Age**	0.154	0–∞	0.960
**Sex**	254.21	0.236–36685	0.964
**Hypertension**	815.33	0–∞	1
**Diabetes**	0.031	0–∞	0.981

## Data Availability

Not applicable.

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
