# Peer review of "Effect of Long-Term Proton Pump Inhibitor Use on Blood Vitamins and Minerals: A Primary Care Setting Study"

_jcm, 2023, doi:10.3390/jcm12082910_

Round 1

Reviewer 1 Report (New Reviewer)

Dear Editor,

The study looked at the effects of long-term use of proton pump inhibitors (PPIs) on micronutrient absorption. The study recruited 66 subjects, with 30 in the PPI group and 36 in the control group. The study found that long-term PPI users had lower red blood cell count but similar hemoglobin levels. No significant differences were found in blood iron, ferritin, vitamin B12, and folate levels. However, vitamin D deficiency was observed more frequently in the PPI group (100%) than in the control group (30%, p<0.001), with lower blood levels in PPI users. No differences in calcium, sodium, and magnesium were observed. PPI users had lower phosphate levels than controls and there was a non-significant trend for zinc deficiency in PPI users. The study concluded that chronic PPI users may encounter alterations in some micronutrients involved in bone mineral homeostasis and that the effect on zinc levels warrants further investigation.

The purpose of this study is to investigate the impact of proton pump inhibitors (PPIs) on the blood levels of various minerals and vitamins. However, the methodology employed in this study is flawed and therefore the conclusions drawn are not valid. The sample size estimation is inadequate and the groups compared are not comparable due to the significant age discrepancy between the treatment and control groups. Additionally, no nutritional assessment was conducted, no consideration was given to background diseases, and no evaluation of other medications or potential interactions was performed. Furthermore, the data presented in Table 1 contradicts established knowledge regarding the influence of PPIs on gastrin levels (2). In order to arrive at valid conclusions, it is essential that the patient characteristics, medical comorbidities, concurrent medications, and nutritional habits be carefully compared between the groups under investigation.

References:

1. Sahota O. Understanding vitamin D deficiency. Age Ageing. 2014 Sep;43(5):589-91. doi: 10.1093/ageing/afu104. Epub 2014 Jul 28. PMID: 25074537; PMCID: PMC414349

2. Helgadottir H, Bjornsson ES. Problems Associated with Deprescribing of Proton Pump Inhibitors. Int J Mol Sci. 2019 Nov 2;20(21):5469. doi: 10.3390/ijms20215469. PMID: 31684070; PMCID: PMC6862638

Author Response

The study looked at the effects of long-term use of proton pump inhibitors (PPIs) on micronutrient absorption. The study recruited 66 subjects, with 30 in the PPI group and 36 in the control group. The study found that long-term PPI users had lower red blood cell count but similar hemoglobin levels. No significant differences were found in blood iron, ferritin, vitamin B12, and folate levels. However, vitamin D deficiency was observed more frequently in the PPI group (100%) than in the control group (30%, p<0.001), with lower blood levels in PPI users. No differences in calcium, sodium, and magnesium were observed. PPI users had lower phosphate levels than controls and there was a non-significant trend for zinc deficiency in PPI users. The study concluded that chronic PPI users may encounter alterations in some micronutrients involved in bone mineral homeostasis and that the effect on zinc levels warrants further investigation.

The purpose of this study is to investigate the impact of proton pump inhibitors (PPIs) on the blood levels of various minerals and vitamins. However, the methodology employed in this study is flawed and therefore the conclusions drawn are not valid. The sample size estimation is inadequate and the groups compared are not comparable due to the significant age discrepancy between the treatment and control groups. Additionally, no nutritional assessment was conducted, no consideration was given to background diseases, and no evaluation of other medications or potential interactions was performed. Furthermore, the data presented in Table 1 contradicts established knowledge regarding the influence of PPIs on gastrin levels (2). In order to arrive at valid conclusions, it is essential that the patient characteristics, medical comorbidities, concurrent medications, and nutritional habits be carefully compared between the groups under investigation.

We agree about the limitations highlighted by the referee regarding sample size, age and nutritional assessment; for these reasons, these aspects have been largely discussed in the manuscript.

About age, we have conformed this as a bias in the “Patient” section. However, considering the need for the study to have exclusion criteria and to recruit only patients with a continuous PPI assumption for more than 12 months, our population represented the only eligible patients in our cohort from a family practitioner. Furthermore, by answering to a request of another reviewer in a previous round of revision, we added a multivariate analysis (Table 2 in the revised manuscript), which did not find any factor associated with vitamin D deficiency, including age.

Furthermore, we stated in the Methods that “We excluded subjects assuming oral or parenteral integrators containing iron, folate, vitamin B12 or D, or assuming other medications (such as corticosteroids) which could influence vitamin and micronutrient levels. Additionally, we ruled out subjects with diseases (atrophic gastritis, celiac disease, inflammatory bowel disease, haematological disease) that could cause alterations in micronutrients blood levels”. On the other, considering the need for the study to have exclusion criteria and to recruit only patients with a continuous PPI assumption for more than 12 months, our population represented the only eligible patients in our cohort from a family practitioner.

Regarding the gastrin levels, we did not find any difference between the two groups because the p was not significant, therefore the fact that levels were higher in controls than in PPI group is not relevant nor statistically significant, as pointed out by the reviewer.

References:

  1. Sahota O. Understanding vitamin D deficiency. Age Ageing. 2014 Sep;43(5):589-91. doi: 10.1093/ageing/afu104. Epub 2014 Jul 28. PMID: 25074537; PMCID: PMC414349
  2. Helgadottir H, Bjornsson ES. Problems Associated with Deprescribing of Proton Pump Inhibitors. Int J Mol Sci. 2019 Nov 2;20(21):5469. doi: 10.3390/ijms20215469. PMID: 31684070; PMCID: PMC6862638

We added the suggested references

Reviewer 2 Report (New Reviewer)

The authors presented “Effect of Long Term Proton Pump Inhibitors Use on Blood Vit-1 amin and Minerals: A Primary Care Setting Study”. However, I have some concerns following below comments.

 Major:

1) Did you include PPI user or control as variables in multivariate analysis of factors associated with Vitamin D deficiency? If not, I think you should do because vitamin D deficiency was more frequently in PPI group than control group.

2)What is the novel findings? You should emphasize the finding at first paragraph in Discussion section. Readers are not interested in what we don't know yet, but are interested in only the novel findings of your study. Discussion section is too long for a few results.

Author Response

The authors presented “Effect of Long Term Proton Pump Inhibitors Use on Blood Vit-1 amin and Minerals: A Primary Care Setting Study”. However, I have some concerns following below comments.

 Major:

  • Did you include PPI user or control as variables in multivariate analysis of factors associated with Vitamin D deficiency? If not, I think you should do because vitamin D deficiency was more frequently in PPI group than control group.

We confirm that the multivariate analysis was conducted only in the PPI group. We are sorry if we did not catch the message of the referee, however, if he/she wants, we can extend the analysis to both groups.

2)What is the novel findings? You should emphasize the finding at first paragraph in Discussion section. Readers are not interested in what we don't know yet, but are interested in only the novel findings of your study. Discussion section is too long for a few results.

The main novelty of this paper is that it was conducted in a primary care (general practitioner) setting. The other novelty regards zinc levels, which has been properly discussed. The discussion is so long because that paper underwent several previous round of revisions; former reviewers asked to discuss and integrate some aspects in the discussion, adding supplementary references. Therefore we believe that a length cut may alter the suggestions of other reviewers.

Round 2

Reviewer 1 Report (New Reviewer)

The purpose of this study is to investigate the impact of proton pump inhibitors (PPIs) on the blood levels of various minerals and vitamins. However, the methodology employed in this study is flawed and therefore the conclusions drawn are not valid. The groups compared are not comparable due to the significant age discrepancy between the treatment and control groups. Additionally, no nutritional assessment was conducted, no consideration was given to background diseases, and no evaluation of other medications or potential interactions was performed. Furthermore, the data presented in Table 1 contradicts established knowledge regarding the influence of PPIs on gastrin levels (1). In order to arrive at valid conclusions, it is essential that the patient characteristics, medical comorbidities, concurrent medications, and nutritional habits be carefully compared between the groups under investigation.

In conclusion: the study methods in this manuscript do not support the conclusions reached. Patients’

characteristics in both groups should be identical, medical comorbidities as well as concomitant

medications and nutrition habits should be compared between the groups.  

REF:

Helgadottir H, Bjornsson ES. Problems Associated with Deprescribing of Proton Pump Inhibitors. Int J Mol Sci. 2019 Nov 2;20(21):5469. doi: 10.3390/ijms20215469. PMID: 31684070; PMCID: PMC6862638

This manuscript is a resubmission of an earlier submission. The following is a list of the peer review reports and author responses from that submission.

Round 1

Reviewer 1 Report

Upon decision of the editor to re-evaluate the manuscript I agree that the scatterplots of the various micronutrients help to understand the results.

I am not happy with the changes to the introduction as the citation about post-menaupausal women dresses a selected group with specific risk factors. On the other hand, e revision on the risk in children (DOI: 10.1097/MPG.0000000000003246) came to different conclusions. Therefore non-directed suggestions would be more appropriate.

It is true that PPI users tend to be older than non-users. However it was the author's selection that came to these groups. It is an undeniable bias in the study. It cannot be dismissed or ignored.

Concordance and discrepancies from similar results still are not properly explained and the number of shortcomings does not favour the present study.

Author Response

Upon decision of the editor to re-evaluate the manuscript I agree that the scatterplots of the various micronutrients help to understand the results.

I am not happy with the changes to the introduction as the citation about post-menaupausal women dresses a selected group with specific risk factors. On the other hand, e revision on the risk in children (DOI: 10.1097/MPG.0000000000003246) came to different conclusions. Therefore non-directed suggestions would be more appropriate.

We replaced the reference 7 with another paper reporting a meta-analysis of 18 studies, which showed that PPI use could moderately increase the risk of hip fracture with a RR = 1.26 (Zhou B, Huang Y, Li H, Sun W, Liu J. Proton-pump inhibitors and risk of fractures: an update meta-analysis. Osteoporos Int. 2016;27(1):339-47). Such article does not address the risk in a specific cohort of patients.

It is true that PPI users tend to be older than non-users. However it was the author's selection that came to these groups. It is an undeniable bias in the study. It cannot be dismissed or ignored.

We have conformed this as a bias in the “Patient” section. However, considering the need for the study to have exclusion criteria and to recruit only patients with a continuous PPI assumption for more than 12 months, our population represented the only eligible patients in our cohort from a family practitioner. Furthermore, by answering to a request of another reviewer, we added a multivariate analysis (Table 2 in the revised manuscript), which did not find any factor associated with vitamin D deficiency, including age.

Concordance and discrepancies from similar results still are not properly explained and the number of shortcomings does not favour the present study.

We agree that there are several discrepancies in literature data. and precisely their heterogeneity about PPI-induced adverse events could justify our findings. However, in agreement to reviewer’s suggestion, we added a further study which found that PPI consumers had slightly higher vitamin B12 (Lerman TT, Cohen E, Sochat T, Goldberg E, Goldberg I, Krause I. Proton pump inhibitor use and its effect on vitamin B12 and homocysteine levels among men and women: A large cross-sectional study. Am J Med Sci. 2022;364(6):746-751), which is in full disagreement with another that did not find any effect on folate, B12, calcium and vitamin D (Hatemi İ, EsatoÄŸlu SN. What is the long term acid inhibitor treatment in gastroesophageal reflux disease? What are the potential problems related to long term acid inhibitor treatment in gastroesophageal reflux disease? How should these cases be followed? Turk J Gastroenterol. 2017;28(Suppl 1):S57-S60).

Reviewer 2 Report

Aim of Losurdo and colleagues was to study the effect of long term proton pump inhibitors on micronutritents, in a primary care setting.

Overall, I think that this paper is well written and the topic of interest. However, I think there are several limitations that lowers the meaningfulness of this study.

- The main problem with this study is methodological: patients under PPI were compared with patients without PPI use. However, the nutritional differences evaluated between these two group  are not significant: the two groups are composed by patients with different characteristics and different nutrient baseline values. Patients using PPI, that are also older than patients not using PPI, may have had different nutrient levels even before the PPI use. The appropriate study model (to evaluate PPI effect of nutrients) would have assessed the nutrient levels before and after PPI use on the same patients. 

- The results report only the sex and age of the patients. Several parameters are not reported, including current diet, current therapies and other comorbidities, which may strongly impact the nutritional profile. 

- There is a lack of multivariate analysis to assess the real impact of other confounding factors on nutrient levels.

- The authors declared they ruled out atrophic gastritis as a confounding factor. However, this is in contrast to table 1, in which 10 patients were positive for APCA, so they have at least a potential atrophic gastritis. 

- The title reported "Proton Pump Inhibitors" but the study evaluated only the effect of pantoprazole. 

Author Response

Aim of Losurdo and colleagues was to study the effect of long term proton pump inhibitors on micronutritents, in a primary care setting.

Overall, I think that this paper is well written and the topic of interest. However, I think there are several limitations that lowers the meaningfulness of this study.

- The main problem with this study is methodological: patients under PPI were compared with patients without PPI use. However, the nutritional differences evaluated between these two group  are not significant: the two groups are composed by patients with different characteristics and different nutrient baseline values. Patients using PPI, that are also older than patients not using PPI, may have had different nutrient levels even before the PPI use. The appropriate study model (to evaluate PPI effect of nutrients) would have assessed the nutrient levels before and after PPI use on the same patients.

We perfectly agree with the comment of the reviewer. Indeed a study with follow up before and after a long term assumption of PPI could have been more proper and given more interesting results. We mentioned this limitation in the Discussion paragraph.

- The results report only the sex and age of the patients. Several parameters are not reported, including current diet, current therapies and other comorbidities, which may strongly impact the nutritional profile.

We agree that the lack of the analysis of the current diet is a major limitation, which has been already considered and discussed in the paper. Regarding comorbidities, the most common ones were hypertension (n=26) and diabetes (n=6), which have been entered a multivariate analysis in table 2, as requested by the reviewer, and were not significantly associated with vitamin D deficit. Furthermore, we stated in the Methods that “We excluded subjects assuming oral or parenteral integrators containing iron, folate, vitamin B12 or D, or assuming other medications (such as corticosteroids) which could influence vitamin and micronutrient levels. Additionally, we ruled out subjects with diseases (atrophic gastritis, celiac disease, inflammatory bowel disease, haematological disease) that could cause alterations in micronutrients blood levels”. On the other, considering the need for the study to have exclusion criteria and to recruit only patients with a continuous PPI assumption for more than 12 months, our population represented the only eligible patients in our cohort from a family practitioner.

- There is a lack of multivariate analysis to assess the real impact of other confounding factors on nutrient levels.

Thank you for the comment. We added a multivariate analysis (Table 2 in the revised manuscript), which did not find any factor associated with vitamin D deficiency.

- The authors declared they ruled out atrophic gastritis as a confounding factor. However, this is in contrast to table 1, in which 10 patients were positive for APCA, so they have at least a potential atrophic gastritis.

Despite the positivity of APCA, all patients underwent upper endoscopy with biopsy sampling, and in no cases atrophic gastritis was diagnosed. Therefore such positivity could be considered as an incidental finding.

- The title reported "Proton Pump Inhibitors" but the study evaluated only the effect of pantoprazole.

In this case, we decided to be generic in the title. Indeed, despite Pantoprazole was prescribed to all patients, the consequences on micronutrient absorption are due to suppression of acid production, which is a class-related (PPI) effect, and it is not related to the single molecule (Pantoprazole).

Round 2

Reviewer 2 Report

I appreciate the corrections made by the Authors. Nevertheless, I believe that the methodological limitations of the study, as expressed in report 1, limit the significance of the results and the scientific value.

Just as a note, I point out that potential autoimmune gastritis is defined as antibody positivity in the absence of lesions at gastroscopy. This condition may evolve to overt atrophy over time and may be associated with nutritional alterations. (see Lenti et al. Time course and risk factors of evolution from potential to overt autoimmune gastritis. 2022) Finally, I insist that, in accordance with the study design, the results obtained cannot be extended to other PPIs. 

Author Response

I appreciate the corrections made by the Authors. Nevertheless, I believe that the methodological limitations of the study, as expressed in report 1, limit the significance of the results and the scientific value.

Just as a note, I point out that potential autoimmune gastritis is defined as antibody positivity in the absence of lesions at gastroscopy. This condition may evolve to overt atrophy over time and may be associated with nutritional alterations. (see Lenti et al. Time course and risk factors of evolution from potential to overt autoimmune gastritis. 2022) Finally, I insist that, in accordance with the study design, the results obtained cannot be extended to other PPIs. 

We discussed the concept of potential autoimmune gastritis and added the suggested reference.

Finally, we replaced several times in the text "PPI" with Pantoprazole
